# OpenReview forum: "Non-deep Networks"
_NeurIPS.cc/2022/Conference — NeurIPS 2022 Accept_

### Official Review · Reviewer_cvGo · 2022-07-10

**Rating:** 7
**Confidence:** 5
**Soundness:** 3 good
**Presentation:** 3 good
**Contribution:** 3 good

**Summary:**

This paper proposes a non-deep network for low latency. The key designs and techniques include multi-stream topology, RepVGG-style structural re-parameterization and SSE block. Reasonable results on ImageNet, CIFAR and COCO are reported. The results show that shallow models can perform competitively with deep ones.

**Questions:**

Suggestions:
1. A discussion with RepVGG may highlight the contributions of this paper as well as show the differences. For example, "this paper focus on the depth while RepVGG was more about the simplicity, .........., this paper is related to RepVGG because a RepVGG-style overall architecture is suitable for shallow model design".

2. L103: "VGG-style block" seems confusing. A reader may question why a single 3x3 conv is called a block. I would suggest the authors use "VGG-style architecture" or "RepVGG-style design".

3. L133: missing reference. L143: missing reference

4. Caption of Table 2: I would suggest the authors use 3$\times$3 instead of 3X3.

5. L311: "consists of consists of"


**Limitations:**

Have the authors adequately addressed the limitations and potential negative societal impact of their work? Yes. I appreciate the discussions.

**Strengths And Weaknesses:**

Strengths:
S1. This paper shows that a 12-layer model can achieve accuracy of over 80% on ImageNet and an AP of 48% on MS-COCO. The results show that CNNs do not need to be very deep and research on shallow models is still useful, which is valuable to the literature. This is the primary reason I recommend accepting this paper.
S2. The key techniques include RepVGG-style structural re-parameterization, multi-stream design and SSE block. The motivations are well explained and the effects are verified.
S3. The results with custom hardware settings are particularly interesting. (For the multi-GPU version, each stream is launched on a separate GPU) The results and discussions of the communication overhead are useful.

Weaknesses
W1. The reported results are all small- and middle-sized models. It is not sure if heavy-weight models show different depth v.s. performance trade-offs. Of course, I understand the authors may have limited computing resources and did not lower my rating for lack of such large-scale experiments.
W2. Though the discussions of the future hardware are interesting, this paper shows no clue on how to realize it. Maybe the authors can show a potential direction.
W3. The writing can be improved. Please see the suggestions below.

---

> ### Author Response · Authors · 2022-08-02
> **Response to Reviewer cvGo**
>
> We thank the reviewer for positive comments and helpful feedback on our work. We address the concerns below:
>
> **Concern: Though the discussions of the future hardware are interesting, this paper shows no clue on how to realize it. Maybe the authors can show a potential direction.**
>
> Thanks for the question! One way to realize hardware to support parallel architectures is to have a multi-die GPU where fast connections can be established between GPUs. In fact, we are seeing a trend towards such hardware with Cerebras releasing WSE-2, an 84 die AI chip with fast interconnect and AMD releasing the first multi-die GPU [1]. Such hardware is emerging because there are constraints with reducing the size of the transistor and increasing the area of ​​the die (more detail in L162-172).
>
> [1] https://www.cerebras.net/blog/an-ai-chip-with-unprecedented-performance-to-do-the-unimaginable/
> [2] https://www.amd.com/en/graphics/instinct-server-accelerators
>
> **Concern: A discussion with RepVGG may highlight the contributions of this paper as well as show the differences. For example, "this paper focuses on the depth while RepVGG was more about the simplicity, .........., this paper is related to RepVGG because a RepVGG-style overall architecture is suitable for shallow model design".**
>
> Thanks for the great suggestion. We will do so.
>
> **Concern: L103: "VGG-style block" seems confusing. I would suggest to use "VGG-style architecture" or "RepVGG-style design".**
>
> This is a great suggestion. We will do so.
>
> **Concern: L133: missing reference. L143: missing reference**
>
> Thanks for pointing this out. We will update.
>
> **Concern: Caption of Table 2: I would suggest the authors use 3×3 instead of 3X3.**
>
> Thanks for the suggestion. We will revise accordingly.
>
> **Concern: L311: "consists of consists of"**
>
> Thanks for pointing this out. We will revise.

---

> ### Comment · Reviewer_cvGo · 2022-08-06
> **Response to the rebuttal**
>
> The authors addressed my concerns, so I decided to keep my rating.

---

### Official Review · Reviewer_zBDf · 2022-07-10

**Rating:** 7
**Confidence:** 4
**Soundness:** 3 good
**Presentation:** 3 good
**Contribution:** 4 excellent

**Summary:**

The presented paper questions the importance of the depth in neural architectures to provide state-of-the-art-results. There are two main motivations for this paper, the first being that very deep networks are inherently ill-suited for real-time systems, since input data needs to be processed sequentially through each layer. The second motivation is that computations within shallow networks using parallel substructures can be parallelized, hence (theoretically) reducing the effective computation time. However, depth being a central feature allowing neural architecture to perform complex classification tasks, the main challenge addressed by this paper is to derive a class of shallow architectures whose performances remain competitive against state-of-the-art deep architecture. The resulting architecture, called ParNet, is rather flexible in the sense that one can scale its representational power by increasing the number of parallel substructure. The authors specifically discuss the importance of certain module with respect to the final performance reported in the experiment section.

**Questions:**

Can the author provide a setting example where they can lower-bound the time it would take for a ResNet-like or VGG-like architecture to process an input, both theoretically and empirically, and show to what extent parallelism gives a crucial advantage ?
Throughout the paper, the authors mention situations where multi-stream shallow networks should provide much faster computations.
For example, table 3 compare ParNet with and without parallelism but do not compare the speed with other architectures.
Furthermore, the authors promote the need for robust shallow architecture to reduce latency in real-time systems such as autonomous vehicles. The closest experiment addressing this issue is the MS-COCO challenge as it is a detection rather than a classification challenge, but parallelism is not employed here, thus resulting in less than 25% improvement in latency while the depth is reduced by 75% compared to the baseline. Can the authors explain why they did not include more experiments on model parallelism while strongly motivating their paper for this particular purpose ?

**Limitations:**

One limitation of the paper, actually discussed by the authors, is that non-deep networks are not a replacement for their deep counterparts. They motivate such claim by saying that non-deep networks still require a high number of parameter and flops. From a more theoretical side, however, the reviewer strongly wonders if there are classes of functions actually requiring a certain depth to be effectively approximated by a neural network. This last point could help identify the use cases where non-deep networks will never be an acceptable replacement for their deep counterparts.

**Strengths And Weaknesses:**

The paper motivates its research effort to derive efficient shallow architectures with several arguments, the first one being that shallow networks are highly desirable in real-time applications. The author also motivates their effort from two "wall-clock" computation arguments, the first one being that deep architectures are hardly biologically plausible given the speed of human reaction with respect to certain stimuli. The authors also try to make reasonable assumptions on the improvement that is likely to be provided by the next generation of hardware, pointing out that neural architectures enabling module parallelism (different from data-parallelism), should benefit from such improvements. Three experiments are conducted to assess the relevance of the architecture:
1) experiments show that ParNet, an architecture of depth only 12, provides performance on par with ResNet-like and VGG-like architectures on CIFAR10, CIFAR100, ImageNet and MS-COCO. The authors also carefully discuss speed versus accuracy in the case of ImageNet.
2) the impact of specific block design and engineering tricks found in the literature with respect to the final performance are discussed to show how they can be employed to reduce depth while maintaining high performance. The claims are backed by an ablation study.
3) rules to increase the statistical capacity are presented, and the increased statistical capacity results in increased performance for ParNet.

Some weaknesses are exposed by the authors themselves. In table 2, RepVGG was found to be faster than ParNet, however authors point out that RepVGG benefits from a highly optimized design which is coherent given that this architecture has been tested and improved on many times over. Another key weakness in the presented paper (not the contribution itself) is the lack of real-world scenario where parallelism is actively employed to reduce computation time, hence showcasing the advantage of ParNet in such setting. Table 3 attempts to show that parallelizing the streams on different GPU does reduce the effective computation time, but it is likely that neither the software nor the hardware are optimized for ParNet paradigm, since they were initially developped to tackle data-parallelism.

---

> ### Author Response · Authors · 2022-08-02
> **Response to Reviewer zBDf**
>
> We thank the reviewer for positive comments and helpful feedback on our work. We address the concerns below:
>
> **Concern: Provide an example where authors can lower-bound the latency of ResNet-like or VGG-like architecture, (theoretically and empirically) and show to what extent parallelism gives a crucial advantage? Table 3 compare ParNet with and without parallelism but do not compare the speed with other architectures.**
>
> Thanks for the interesting question.
>
> Following is a empirical scenario for autonomous vehicles. Ideally, the latency of the autonomy system should be less than or equal to the latency of the sensor captur system. The autonomy systems includes image preprocessing, network prediction and control. Hence, the latency requirement for the network prediction are more stringent. To cater to this requirement, autonomy systems have proposed to use two different detectors operating at different frequencies:  a less accurate detector with high speed and a more accurate detector at low speed [1]. This is because the current detectors based on ResNet-like or VGG-like architecture might be insufficient for accurate prediction at high speed. We believe that parallelism could provide a crucial advantage here by allowing us to operate accurate and fast detectors.
>
> For theoretical lower bound, if we assume a clock speed of 1000 MHz (typical for GPUs), and that each layer can be executed in 10 clock cycles, then a network with 100 Layers can be best executed in 1 ms. With parallelism, and non-deep networks, one can theoretically do the same operation in 0.1 ms assuming a depth of 10.
>
> We provide comparison of speed with other architectures in Table 2.
>
> [1] Shen, A., Tesla Inc, 2020. Machine learning models operating at different frequencies for autonomous vehicles.
>
> **Concern: Why is model parallelism not used for object detection in MSCOCO?**
>
> We found in our experiments that parallelization does not offer much advantage in the object detection case due to the high overhead of transferring the large features of higher resolution. In this case, communication becomes a major bottleneck. We believe that parallelism will be more relevant for this purpose when that issue is reduced i.e. when the relative data transfer latency will be less compared to the execution time of model layers. Developments like multi-die GPU are steps in this direction.
>
> **Concern: From a theoretical side, are there classes of functions actually requiring a certain depth to be effectively approximated by a neural network. This could help identify the use cases where non-deep networks will never be an acceptable replacement for their deep counterparts.**
>
> Thanks for the interesting question! In fact, the classic work by Cybenko et al. [1] shows that even a single layer neural network, when sufficiently wide, can approximate any function with arbitrarily small error. However, such a network might be impractical because they may need large number of parameters to satisfy the width requirement. In our work, we show how one can use a non-deep network with reasonable parameter count for computer vision applications. We hope our work can inspire theoretical investigation in this direction.
>
> [1] Approximation by Superpositions of a Sigmoidal Function

---

### Official Review · Reviewer_t5as · 2022-07-11

**Rating:** 7
**Confidence:** 4
**Soundness:** 4 excellent
**Presentation:** 3 good
**Contribution:** 3 good

**Summary:**

This paper presents a new CNN architecture designed for the purpose of having a relatively low depth (12 layers) while still providing results on par with (deeper) classic ResNets on various vision benchmarks, demonstrating that there may be other mechanisms than depth at play to obtain state of the art results with CNNs. The strong expressive power of the architecture is obtained thanks to parallel substructures in the network, each operating at different scale, then fusing together their information content before the final layers. In-depth ablations study are done to demonstrate the relevance of many components of the design.
This work is motivated by the goal of decreasing latency, which can be impacted by depth. In order to improve this metrics, it is shown that, once trained, the network can be expressed equivalently as a classic single stream CNN, increasing its speed at inference.

**Questions:**

* Why/How did you chose the depth of 12 ?
* What is the effect of the depth in your new architecture ?

**Limitations:**

Yes.

**Strengths And Weaknesses:**

$\textbf{Strengths}$

The paper is clearly motivated and very well written. The experiments are numerous, thoroughly comparing the newly introduced CNN architecture to deeper ones on several classic vision datasets. An in-depth ablation study quantifies the impact of most parts of the new design, reporting impressive results for a 12 layers deep network. Comparison between this new design and other standard architectures in terms of latency is done, showing promising results for ParNet.

$\textbf{Weaknesses}$

While the effect of increasing the number of parameters and the number of streams have been studied on this new architectures, the effect of the depth is suprisingly not shown while being the highlighted metric of this paper. How ParNets with a number of layers of similar order of magnitude (e.g., 10,11,13 or 14) perform ? If depth/latency was not an issue, would scaling ParNet to greater depths further increase its performances on standard benchmarks ?

---

> ### Author Response · Authors · 2022-08-02
> **Response to Reviewer t5as**
>
> We thank the reviewer for positive comments and helpful feedback on our work. We address the concerns below:
>
> **Concern:  What is the effect of changing depth on ParNet. Would scaling ParNet to greater depths further increase its performances?**
>
> We thank the reviewer for the suggestion. Following we show results with ParNet variants with depth 9, 12 and 15. M variants have 128, 256 and 512 channels; L have 160, 320 and 640 channels; XL have 200, 400 and 800 channels in the three branches (Fig. 1).
>
> | Size | Depth | Latency (in ms) | ImageNet Top-1 Acc |
> |--:|--:|--:|--:|
> | M | 9 |  3.1 | 73.9 |
> | L | 9 |  3.2 | 75.1 |
> | XL | 9 |  3.7 | 76.1 |
> | M | 12 |  3.8 | 76.6 |
> | L | 12 |  4.0 | 77.7 |
> | XL | 12 |  4.4 | 78.6 |
> | M | 15 |  4.8 | 77.0 |
> | L | 15 |  4.9 | 78.4 |
> | XL | 15 |  5.4 | 79.4|
>
>
> We find that decreasing the depth of ParNet from 12 to 9 reduces latency but also reduces performance. ParNet-XL with depth 9 and ParNet-M with depth 12 have similar latency (3.7 vs 3.8 ms) but slightly worse performance (76.1 vs 76.6).
>
> Similarly, increasing the depth of ParNet from 12 to 15 increases performance but also increases latency. ParNet-M with depth 15 is both slower and less accurate than ParNet-L with depth 12.
>
> **Concern: Why/How did you choose the depth of 12?**
> As shown in the previous table, we find the depth of 12 to be near optimal for ParNet when considering the trade-off between latency and accuracy. Hence, we choose a depth of 12.

---

### Official Review · Reviewer_momd · 2022-07-12

**Rating:** 4
**Confidence:** 4
**Soundness:** 1 poor
**Presentation:** 2 fair
**Contribution:** 3 good

**Summary:**

Prevailing wisdom suggests neural networks must be of sufficient depth in order to achieve competitive or useful accuracy.  The authors challenge this belief and design a family of networks called ParNets, which use parallel structures that fork quickly from the input and converge again close to the output.  ParNets are built of ParNet Blocks, made up of Rep-VGG blocks with Skip-SE layers to increase the receptive field and avoid increasing depth.  The authors present results on a variety of tasks and against a variety of baseline networks.  Generally, ParNets are able to achieve competitive accuracy and latency.  Ablation studies show the impact of several strategies for boosting accuracy and how scaling various dimensions of ParNets (input resolution, number of streams, number of parameters) affects their accuracy.


**Questions:**

What happens if the streams of ParNet are made deeper?
Conversely, what if depth is decreased - is there a minimum necessary depth for ParNet?

Am I missing a fundamental relationship between depth and latency that doesn't exist between other axes, like layer width, and latency?  (Formulating an equation for minimum latency seems like it should be based on FLOP count and bandwidth requirements rather than something more abstract like "depth.")


**Limitations:**

Yes, the limitations have been discussed adequately.

**Strengths And Weaknesses:**

== Strengths ==

The submission is intriguing - it boldly challenges conventional wisdom and shows that networks need not be very deep at all to achieve useful, much less competitive, accuracy.  I found the network design methodology to be interesting and grounded in reason.  I imagine much of the NeurIPS attendees and readership will find the topic and approach compelling, at least from a theoretical standpoint.

The text is not hard to follow; I rarely found myself needing to re-read a sentence due to having trouble parsing it or placing it into context with the preceding text.

The breadth of experiments is appreciated: not only are there accuracy results, but we also see latency for some experiments, comparisons with different baselines, ablations studies of all types, and different tasks are represented.

Finally, the ultimate goal of showing that a shallow network can compete with deep networks is successful: the accuracy of the 12-layer networks, regardless of the number of streams, stream widths, or input size, is sufficient to make the field take notice.

== Weaknesses ==

For the successes I describe above, I think the exposition and successive analysis are in need of attention before I can recommend the submission for acceptance.

I found the motivation to be lacking nuance and the focus on network depth to be myopic.
- In the abstract, it is asserted that "latency is fundamentally dependent on the depth of the network."  This is also true of the number of parameters width of each layer - wider layers require more FLOPs, increasing latency on a given piece of hardware.  Similarly, latency is fundamentally dependent on the input resolution, choice of activation function (is it a simple clamp() function, like ReLU, or does it involve a transcendental function?), and choice of layer types (3x3 convolution vs. 1x1 convolution vs. fully-connected vs. depthwise vs. …).
- This focus on depth gets stranger with line 20's "the lowest achievable latency is d/f" - layer-seconds per cycle is an odd unit of latency.  I think I understand the intent, but I can't help but feel it could be stated less sensationally.
- The focus on depth is similarly out of place in Section 3.5's Line 177-178: "All these factors make non-deep parallel structures advantageous…", but nothing in this section was related to network depth, just parallel operations.
- Line 264 suggests that "it is surprising that a mere depth-12 network could achieve…. This further indicates that non-deep networks can work as well as deeper counterparts."  Strictly speaking, any surprise at the results does not indicate anything about non-deep networks' relative performance to their deeper counterparts.  What's really shown is that ParNet's structure allows for shallower networks than the other network structures studied.

An obvious experiment was left un-performed, in light of the claim that non-deep networks can work as well as deeper counterparts: what happens if the streams of ParNet are made deeper?  Does performance not improve, supporting this assertion?  Conversely, what if depth is decreased - is there a minimum necessary depth for ParNet?  Any results here would not diminish the contributions, but they would help put them into perspective.

I noted several unfair or misleading comparisons in the results:
- Figure 1 mentions that "for fairness," the authors "exclude results with longer training, higher resolution, or multi-crop testing."  However, the authors also seem to report their own networks' accuracy results with higher resolution in Table 4's results, as described in Line 237: "We use this higher image resolution for ParNext-XL and ParNet-XL-CSP."
- Table 2 presents latency results for several baselines networks and several ParNets, with the stated conclusion that, "In spite of communication overhead, ParNet is faster than similar-performing ResNets."  This is not a fair comparison, as the ParNets are using 3x as many compute resources, detailed in Line 210-211: "… for the multi-GPU version, we use 3 GPUs."  This detail is not reflected in the table, where most readers would notice.
- Table 6 is organized such that models with similar sizes are grouped together, and then ParNets are compared only within those groups - this shows that ParNets are within the top three most accurate networks in each group.  However, this disguises other potential comparisons, such as DenseNet (Bottleneck+Compression) with a depth of 250, with the final group.  If it were included, it would show that a network with less than half as many parameters has higher accuracy than the largest ParNet model.

I was waiting for a comparison with ensembling, which is an obvious cousin to the ParNet structure: send an input to separate networks, collect the results, and reduce to a single output.  However, the only comparison is with ensembling ParNet-M models.  A better choice might be ResNet-50; as presented in Table 2, it has higher accuracy, fewer parameters, and fewer FLOPs than ParNet-M.  Ensembling two RN50 models would have roughly the same number of parameters as a single ParNet-L.  Three RN50s would make use of the three GPUs afforded to ParNet-L in Table 2, so latency for either ensemble would be roughly identical.  If the latency is roughly the same for improved accuracy (pending the experiment!), then why would ParNets have an advantage?

The conclusion from Figure 3 on Lines 299-301 is that "Based on these charts, we see no saturation in performance while scaling ParNets.  This indicates that by increasing compute, one could achieve even higher performance with ParNet while maintaining low depth."  This is in contrast to Table 10, though, which shows explicitly that scaling from a third to a fourth branch reduces Top-1 accuracy.  It seems that Figure 3 simply does not scale far enough to see the saturation shown in Table 10.

---

> ### Author Response · Authors · 2022-08-02
> **Response to Reviewer momd (1/2)**
>
> We thank the reviewer for positive comments and helpful feedback on our work. We address the concerns below:
>
> **Concern: The focus on depth is myopic. It is asserted that “latency is fundamentally dependent on the depth of the network." This is also true of other factors such as the number of parameters, the input resolution, choice of activation function, and choice of layer types. Am I missing a fundamental relationship between depth and latency that doesn't exist between other axes, like layer width, and latency?**
>
> Thanks for pointing out the confusion, We would like to clarify a potential misunderstanding. It is correct, that for a particular piece of general purpose hardware like GPU, latency is dependent on all the factors mentioned by the reviewer like number of parameters, width of layer etc. This is because the current GPUs have limited memory and number of cores. Hence, the extent of parallelization is not perfect.
>
> But in the context of our work (L20 - L22), we are referring to the lowest achievable latency for a network with optimal hardware for parallelization. One way to achieve this latency would be to  print the entire network on a chip. Further, future hardware with more memory and cores would also facilitate such parallelization. For better clarity, we will rephrase to “Lowest theoretical latency is dependent on depth” We are open to suggestions from the reviewer.
>
>
> **Concern: The focus on depth gets stranger with line 20's "the lowest achievable latency is d/f" - layer-seconds per cycle is an odd unit of latency. I understand the intent, but feel it could be stated less sensationally.**
>
> Our intention was not to sensationalize but to make a point about a theoretical bound on latency. In hindsight, we feel a theoretical limitation might be a better way to say it. We are open to suggestions from the reviewer.
>
> **Concern: The focus on depth is out of place in Section 3.5's Line 177-178: "All these factors make non-deep parallel structures advantageous…". Nothing in this section was related to network depth, just parallel operations.**
>
> We regret the confusion and would like to clarify the misunderstanding. Depth matters in this section because we discuss here that one cannot increase the processor frequency because of physical limitations. Hence, networks with less sequential operations (like non-deep networks) are advantageous. We will clarify this further in the paper.
>
> **Concern: L264 suggests that "it is surprising that a mere depth-12 network could achieve…. This further indicates that non-deep networks can work as well as deeper counterparts." Strictly speaking, any surprise at the results does not indicate non-deep networks' relative performance to their deeper counterparts. What it shows is that ParNet's structure allows for shallower networks than the other network structures.**
>
> Thanks for the great suggestion! We will revise the statement to the following for clarity: “This further indicates that a non-deep network such as ParNet can work as well as deeper SOTA networks such as ResNet”
>
> **Concern: What happens if the streams of ParNet are made deeper? Does performance not improve, supporting this assertion? Conversely, what if depth is decreased - is there a minimum necessary depth for ParNet? Any results would not diminish the contributions, but help put them into perspective.**
>
>
> We thank the reviewer for the suggestion. Below we show results with ParNet variants with depth 9, 12 and 15. Note that networks with size M have 128, 256 and 512 channels; size L have 160, 320 and 640 channels; size XL have 200, 400 and 800 channels in the three branches (Fig. 1).
>
> We find that decreasing the depth of ParNet from 12 to 9 reduces latency but also reduces performance. ParNet-XL with depth 9 and ParNet-M with depth 12 have similar latency (3.7 vs 3.8 ms) but slightly worse performance (76.1 vs 76.6).
>
> Similarly, increasing the depth of ParNet from 12 to 15 increases performance but also increases latency. ParNet-M with depth 15 is both slower and less accurate than ParNet-L with depth 12.
>
> Overall, we find that the depth of 12 to be better for ParNet when considering both the latency and accuracy.
>
> | Size | Depth | Latency (in ms) | ImageNet Top-1 Acc |
> |--:|--:|--:|--:|
> | M | 9 |  3.1 | 73.9 |
> | L | 9 |  3.2 | 75.1 |
> | XL | 9 |  3.7 | 76.1 |
> | M | 12 |  3.8 | 76.6 |
> | L | 12 |  4.0 | 77.7 |
> | XL | 12 |  4.4 | 78.6 |
> | M | 15 |  4.8 | 77.0 |
> | L | 15 |  4.9 | 78.4 |
> | XL | 15 |  5.4 | 79.4|

---

> > ### Author Response · Authors · 2022-08-02
> > **Response to Reviewer momd (2/2)**
> >
> > **Concern: Figure 1 mentions that results with longer training, higher resolution, or multi-crop testing are excluded for fairness. However, the authors also report ParNet’s accuracy with higher resolution in Table 4's results (L226).**
> >
> > We would like to clarify that there is no contradiction between the two statements. The two statements are being made in different contexts.  Figure 1 reports depth vs accuracy on ImageNet. Auxiliary factors like multi-crop testing and higher resolution can affect the performance of all networks, including ParNet. Hence, for fair comparison we report numbers for all networks (including ParNet) without these auxiliary factors. In Table 5, we separately show the effect of auxiliary factors like longer training, higher resolution and multi-crop testing for ParNet on ImageNet.
> >
> > On the other hand, L226-L240 describes latency vs performance on MSCOCO object detection. We show that within the latency budget, ParNet can use higher resolution and perform better than baseline.
> >
> >
> > **Concern: Table 2 presents latency results with the conclusion that, "In spite of communication overhead, ParNet is faster than similar-performing ResNets." However, ParNets are using 3x as many compute resources (L210-211). This detail should be reflected in the table.**
> >
> > Thanks for the suggestion. We will revise the table caption to make it clear that ParNet uses 3 GPUs.
> >
> > **Concern: Table 6 is organized such that models with similar sizes are grouped together, and then ParNets are compared only within those groups. This disguises other potential comparisons, such as DenseNet (Bottleneck+Compression) with a depth of 250, with the final group. If it were included, it would show that a network with less than half as many parameters has higher accuracy than the largest ParNet model.**
> >
> > Organizing models by size is a well-adopted and clear way to present such results. We are open to suggestions that the reviewer might have. In our opinion, the organization shows how ParNet performs similar to DenseNet and worse than DenseNet (Bottleneck + Comparison) in terms of number of parameters vs accuracy. We also state the same in L257-263.
> >
> > **Concern: Ensemble is an obvious cousin of ParNet. The only comparison is with ensembling ParNet-M models. A better choice might be ResNet-50; as presented in Table 2, it has higher accuracy, fewer parameters, and fewer FLOPs than ParNet-M. If the latency is roughly the same for improved accuracy, then what advantage do ParNets have?**
> >
> > We want to clarify a potential misunderstanding. The model used for ensemble is not ParNet-M but a single-stream ParNet model (Table 9). Ensemble on a single-stream ParNet model allows us to compare the advantage of ensemble vs multi-branches while controlling for other factors like the block structure and depth. Hence, only for ParNet we show the advantage of multiple branches vs ensembles.
> >
> > Regarding comparison with ensembles of deeper networks such as RN50, we agree that ParNets currently are not a replacement for them (L308). As pointed out by the reviewer, our objective is to show that it is possible to build high-performing non-deep networks.
> >
> > **Concern: The conclusion from Fig. 3 (L299-301) is that by increasing compute, one could achieve even higher performance with ParNet while maintaining low depth. This is in contrast to Table 10, though, which shows that scaling from 3 to 4 branches reduces Top-1 accuracy. Figure 3 does not scale far enough to see the saturation shown in Table 10.**
> >
> > Thanks for the suggestion. We want to clarify a confusion. In Table 10, we increase the number of streams while keeping the number of parameters the same. For Figure 3, we mean scaling via increasing compute that can involve increasing the number of parameters and flops. That being said, we agree that there could be saturation beyond the range we tested, so we will revise to clarify it.

---

> > > ### Comment · Reviewer_momd · 2022-08-05
> > > **Comparisons, experiments, and conclusions**
> > >
> > > I still find that the experiments and results require attention in order to support the conclusions as they are presented, or need clarification to avoid incorrect conclusions.
> > >
> > > **[Table 4] We would like to clarify that there is no contradiction between the two statements. ... We show that within the latency budget, ParNet can use higher resolution and perform better than baseline.**
> > >
> > > My concern here is that your first figure sets the tone that all the results use the exact same configuration for fair accuracy comparisons.  Table 4 presents results that do *not* use the same configuration (the ParNet results use a higher resolution), and it is only with careful reading of the text that this becomes clear.  The baseline results' accuracies could be higher under the same experimental configuration.  I understand that their latencies would also increase, and that your results are already better in both metrics (accuracy and latency).  An easy solution is to expand the table to include the ParNet results at the same resolution - this makes the configurations crystal clear and gives the reader some extra information about how ParNets scale.
> > >
> > > **We will revise the table caption to make it clear that ParNet uses 3 GPUs.**
> > >
> > > In addition to clarifying that three GPUs were used, the claim should be tempered with "… faster than similar-performing ResNets when they are not parallelized beyond a single GPU."  The structure of ResNets may not be as trivially parallelizable as ParNets, but it is not impossible, especially for large input sizes typically used for detection tasks (see e.g. "Spatially Parallel Convolutions," Jin et al., ICLR 2018 Workshops, or ensembling [more below]).  Claiming speed superiority when using different computing resources is not fair.
> > >
> > > **Organizing models by size is a well-adopted and clear way to present such results. We are open to suggestions that the reviewer might have. In our opinion, the organization shows how ParNet performs similar to DenseNet and worse than DenseNet (Bottleneck + Comparison) in terms of number of parameters vs accuracy. We also state the same in L257-263.**
> > >
> > > I agree - this organization is commonly-used.  Further, there's certainly no optimal organization for all situations.  There may not be an optimal organization even for just this table!  However, the formatting added to identify the second and third best performing models in each category makes it seem that, for a given parameter limit, ParNet is in the top three networks for CIFAR10 at the upper end of accuracy.  This conclusion would be incorrect, however, when considering all the models that fit into the same parameter constraint.  Either the table and its formatting should not lead a casual reader to an incorrect conclusion, or the proper conclusion should be explicitly mentioned in the text.  In this case, removing the formatting would be sufficient to avoid the implicit suggestion that ParNet is in the top-3 for high-accuracy CIFAR10 models, but I'm curious about what conclusions would fall out if the table were ordered by accuracy, instead. (Either data set; hopefully the other data set's accuracies will be close to ordered.)
> > >
> > > **We want to clarify a potential misunderstanding. The model used for ensemble is not ParNet-M but a single-stream ParNet model (Table 9). ... Regarding comparison with ensembles of deeper networks such as RN50, we agree that ParNets currently are not a replacement for them (L308). As pointed out by the reviewer, our objective is to show that it is possible to build high-performing non-deep networks.**
> > >
> > > Thank you for clarifying my misunderstanding with the ensembled model.  That is not my concern with this experiment, though.  The section's title and text suggest the comparison is between "ParNet vs. Ensembles," but a more accurate description of the experiment might be "Ensembled single-stream ParNet vs. multi-branch ParNet."  I'd much prefer to see the experiment suggested by the original text!
> > >
> > > L308 further qualifies your above statement "... non-deep networks are not a replacement for their deep counterparts *[in] low-compute settings requiring small number of parameter and flops.*"  Ensembling something like RN50 would address a different setting, in which large amounts of parameters and flops are allowed.  If they are not a replacement there, either, then where are they best used?

---

> > > > ### Author Response · Authors · 2022-08-09
> > > > **Response to Reviewer (2/2)**
> > > >
> > > > **Reviewer momd: My concern here is that your first figure sets the tone that all the results use the exact same configuration for fair accuracy comparisons. Table 4 presents results that do not use the same configuration (the ParNet results use a higher resolution), and it is only with careful reading of the text that this becomes clear. The baseline results' accuracies could be higher under the same experimental configuration. I understand that their latencies would also increase, and that your results are already better in both metrics (accuracy and latency). An easy solution is to expand the table to include the ParNet results at the same resolution - this makes the configurations crystal clear and gives the reader some extra information about how ParNets scale.**
> > > >
> > > > Response: As suggested, we will also add result at the same resolution for clarity.
> > > >
> > > > **Reviewer momd: In addition to clarifying that three GPUs were used, the claim should be tempered with "… faster than similar-performing ResNets when they are not parallelized beyond a single GPU." The structure of ResNets may not be as trivially parallelizable as ParNets, but it is not impossible, especially for large input sizes typically used for detection tasks (see e.g. "Spatially Parallel Convolutions," Jin et al., ICLR 2018 Workshops, or ensembling [more below]). Claiming speed superiority when using different computing resources is not fair. **
> > > >
> > > > Response: Thanks for the suggestion. We will update the claim as suggested for clarity.
> > > >
> > > >
> > > >
> > > > **Reviewer momd: I agree - this organization is commonly-used. Further, there's certainly no optimal organization for all situations. There may not be an optimal organization even for just this table! However, the formatting added to identify the second and third best performing models in each category makes it seem that, for a given parameter limit, ParNet is in the top three networks for CIFAR10 at the upper end of accuracy. This conclusion would be incorrect, however, when considering all the models that fit into the same parameter constraint. Either the table and its formatting should not lead a casual reader to an incorrect conclusion, or the proper conclusion should be explicitly mentioned in the text. In this case, removing the formatting would be sufficient to avoid the implicit suggestion that ParNet is in the top-3 for high-accuracy CIFAR10 models, but I'm curious about what conclusions would fall out if the table were ordered by accuracy, instead. (Either data set; hopefully the other data set's accuracies will be close to ordered.)**
> > > >
> > > > Response: As suggested by the reviewer, we will remove the formatting. But we do not believe it is an issue as the table leads to the same conclusion with or without the formatting. On CIFAR-10 and CIFAR-100, ParNet performs better or as well as ResNet and Wide-ResNet; similar to vanilla DenseNet; and worse than DenseNet with compression and bottleneck. ParNets are within the top 3, considering all model classes, which are ResNets and its variants, Wide-ResNets, vanilla DenseNets, DenseNets with compression and bottleneck and ParNets.
> > > >
> > > > **Reviewer momd: Thank you for clarifying my misunderstanding with the ensembled model. That is not my concern with this experiment, though. The section's title and text suggest the comparison is between "ParNet vs. Ensembles," but a more accurate description of the experiment might be "Ensembled single-stream ParNet vs. multi-branch ParNet." I'd much prefer to see the experiment suggested by the original text!
> > > >
> > > > L308 further qualifies your above statement "... non-deep networks are not a replacement for their deep counterparts [in] low-compute settings requiring small number of parameter and flops." Ensembling something like RN50 would address a different setting, in which large amounts of parameters and flops are allowed. If they are not a replacement there, either, then where are they best used?**
> > > >
> > > > Response: Thanks. We will update the title to  "Ensembled single-stream ParNet vs. multi-branch ParNet." as suggested for clarity.
> > > >
> > > > ParNets would be best used with hardware with more parallelization and memory. We will clarify the same in the paper.

---

> > > > > ### Comment · Reviewer_momd · 2022-08-09
> > > > > **ParNet is not demonstrably superior to existing architectures**
> > > > >
> > > > > **As suggested, we will also add result at the same resolution for clarity.**
> > > > >
> > > > > Thank you; could you share the result here so that I may put the current result into context?  Do you also have results for YOLOv4-CSP at the higher resolution for a complete comparison?  What are the two resolutions in question?
> > > > >
> > > > > **The table leads to the same conclusion with or without the formatting.**
> > > > >
> > > > > The formatting in Table 6 leads me to believe that the largest ParNet model is the third-best performer among the models on CIFAR10, since it is underlined and in the largest (implicitly best-performing) group.  This is not an accurate reflection of the data in the full table.  Really, though, this is a small issue in the larger scheme of things.
> > > > >
> > > > >
> > > > > ## My current summary
> > > > >
> > > > > As I've maintained, high network performance (accuracy, not latency/throughput) with so few layers is new and interesting.  However, the motivation is poor, results are confusingly-presented, and the overall importance is questionable:
> > > > >
> > > > > - The motivation for low-depth networks seems to be minimizing latency, which is theoretically bounded by the longest sequential path through a network's operations.  In practice, though, depth is only one of dozens of metrics that dictate a network's latency, on any processor. **It remains unclear to me why low network depth is inherently Good.**
> > > > >
> > > > > - ImageNet results
> > > > >
> > > > >     - ParNet-L's accuracy (77.66/93.60) is roughly the same as ResNet-50 (77.53/93.87), but it has twice as many parameters (54.9M vs. 25.6M) and three times as many FLOPs (26.7B vs. 8.2B).
> > > > >
> > > > >     - As such, it takes 3x the compute resources to roughly match the speed (249 vs. 222 samp/sec).
> > > > >
> > > > >     - The claim in the caption is "In spite of communication overhead, ParNet is faster than similar-performing ResNets."  Another different way to position this result would be to say "thanks to the extra computing resources given to ParNet, it is faster than similar-peforming ResNets."  **In this light, is the result really surprising or impressive?**
> > > > >
> > > > > - CIFAR10/100 results
> > > > >
> > > > >     - The claim in Table 6 is "ParNet performs competitively with deep state-of-the-art architectures while having a much smaller depth."  If we state it another way: "ParNet requires more than twice the number of parameters as  SOTA architectures in order to approach their performance."  **Without a good reason to prefer lower network depth, is this result compelling?**
> > > > >
> > > > > - MSCOCO results
> > > > >
> > > > >     - The authors report being both faster and more accurate than the baseline YOLOv4-CSP.
> > > > >
> > > > >     - However, this comparison is between the baseline at a low resolution and the ParNet at a higher resolution.
> > > > >
> > > > >         - **Without the full results of both networks at both resolutions, it cannot be concluded that that ParNets are superior.**  (Does the baseline network's mAP surpass that of ParNet at the higher resolution?  Did ParNet require higher resolution to meet the baseline mAP?)
> > > > >
> > > > >         - The authors' claim is simply that "non-deep networks can be used as backbones for fast and accurate object detection systems," which seems to be reasonable (pending the actual resolution used).  The **emphasized** note directly above is important in the context of my overall conclusion about the submission, below.
> > > > >
> > > > > - Other parallelization techniques
> > > > >
> > > > >     - ParNets are a natural way to parallelize a network across multiple processors in order to improve accuracy with a given latency budget.
> > > > >
> > > > >     - Ensembling is an existing way to parallelize a network across multiple processors in order to improve accuracy.  **The experiment performed does not provide information about how a ensembling a non-ParNet architecture compares with the ParNet approach.**
> > > > >
> > > > >     - **Similarly, spatially parallel convolutions have not been considered as way to parallelize (for throughput/latency) non-ParNet architectures.**
> > > > >
> > > > >     - As such, the reader cannot conclude anything about the relative quality of the ParNet approach to parallelization.
> > > > >
> > > > > The authors have found a way to pack complexity into each logical layer in order to reduce network depth and to structure the work in each layer in a way that makes it simple to parallelize across multiple processors.  Despite the authors' motivation and claims, it has not been shown that this leads to a demonstrably superior model.  Making a better case for the importance of low network depth, or showing that this method of parallelizing a network (of any depth) is superior to other approaches would make the submission much stronger and improve the importance of its findings.

---

> > ### Comment · Reviewer_momd · 2022-08-05
> > **Remaining doubts about the importance of depth**
> >
> > I truly appreciate the authors' thoughtful responses to all the reviews.  I'll respond to the first half of their response to my review here, and to the second half under that response.
> >
> > To reiterate: such shallow networks performing as well as they do is impressive.  I do not know if the motivation of reducing theoretical latency is particularly compelling, though.  Let me explain by clarifying some of my earlier points in context of your response.
> >
> >
> > **For better clarity, we will rephrase to “Lowest theoretical latency is dependent on depth” We are open to suggestions from the reviewer.**
> >
> > This is a better phrasing, but I would add that this focus on depth as the limiter of latency is dependent on future hardware which is unencumbered by FLOPs or bandwidth.  (If the hardware *could* be limited by these other factors, then disregarding them presents an incomplete view of the lowest theoretical latency.)
> >
> > **In hindsight, we feel a theoretical limitation might be a better way to say it. We are open to suggestions from the reviewer.**
> >
> > To make sure I understand the focus on depth and latency: the idea is that there will be a material difference between a latency of O(10) cycles and a latency of O(50) cycles at a reasonable clock speed?  Even at just 10MHz, the difference between the theoretical lowest latencies is 4us, which is a drop in the bucket of full system latencies which are typically three (or more) orders of magnitude larger, even for low-latency applications.  While I agree with the statement that latency is theoretically bounded by the longest sequential path through a network, I remain unconvinced that this matters in practice.  This is the source of my view that the motivation was a weakness of the submission.
> >
> > **Depth matters in this section because we discuss here that one cannot increase the processor frequency because of physical limitations. Hence, networks with less sequential operations (like non-deep networks) are advantageous. We will clarify this further in the paper.**
> >
> > The conclusion from the observations in this section is *not* that shallow networks are preferable; it's that parallelization matters.  This provides the motivation to parallelize the structure of ParNets - with some math to perform, the way to improve performance when the clock speed cannot be increased is to do them at the same time using more resources.  This technique applies equally to shallow and deep networks.  Suggesting that non-deep networks are advantageous at the end of this section relies on the same assertion of theoretical minimum latency as discussed directly above in our responses, and claiming it here conflates low depth and high parallelism.
> >
> > **We will revise the statement to the following for clarity: “This further indicates that a non-deep network such as ParNet can work as well as deeper SOTA networks such as ResNet”**
> >
> > Table 6 suggests that ResNet may not be SOTA, so I would omit that qualifier.  The best-performing networks in Table 6 are DenseNets; even if they are SOTA, I don't think that ParNet works "as well" given the differences in accuracies.
> >
> > **Overall, we find that the depth of 12 to be better for ParNet when considering both the latency and accuracy.**
> >
> > This is interesting data!  Please include it in the paper; I believe readers will appreciate it.

---

> > > ### Author Response · Authors · 2022-08-09
> > > **Response to Reviewer (1/2)**
> > >
> > > We thank the reviewer for the constructive suggestions. We appreciate that the reviewer has reiterated that the performance of ParNet is impressive considering their depth. Following we have tried to address their concerns:
> > >
> > > **Reviewer momd: “This is a better phrasing, but I would add that this focus on depth as the limiter of latency is dependent on future hardware which is unencumbered by FLOPs or bandwidth.”**
> > >
> > > Response: Thanks for the suggestion! We will add it for clarification.
> > >
> > > **Reviewer momd: “the idea is that there will be a material difference between a latency of O(10) cycles and a latency of O(50) cycles at a reasonable clock speed? Even at just 10MHz, the difference between the theoretical lowest latencies is 4us, which is a drop in the bucket of full system latencies which are typically three (or more) orders of magnitude larger, even for low-latency applications. While I agree with the statement that latency is theoretically bounded by the longest sequential path through a network, I remain unconvinced that this matters in practice. This is the source of my view that the motivation was a weakness of the submission.”**
> > >
> > > Response: There are some issues with this calculation. It assumes that each layer is executed in one GPU cycle. However, each layer may take multiple instruction cycles for memory access, writing result, synchronization etc. Hence, if each layer takes 10 cycles, the theoretical lowest latency difference for the above case would be 40 us. We will clarify it in the paper.
> > >
> > > Also, let's say the number of cores required for perfect parallelism is x. But when a GPU has x/10 cores, then the theoretical maximum latency would be 400 us or .4ms. So the theoretical latency could still be useful with GPUs with relatively larger numbers of cores.
> > >
> > > Further, many applications like robotics and self-driving cars might require inference from multiple networks in real-time. Hence, the latency requirement for each network becomes stringent.
> > >
> > > For some applications, these latency differences might be tolerable. However, they quickly become a limiting factor in some other applications. For example, the control system for Fusion Reactors operates at 10KHz, or 0.1ms [1]. Such constraints restrict the neural network to be small and only of depth 4. Hence, we believe with future applications and hardware, depth and latency trade off will become more important.
> > >
> > > [1] Magnetic control of tokamak plasmas through deep reinforcement learning, Nature 2021
> > >
> > > **Reviewer momd: The conclusion from the observations in this section is not that shallow networks are preferable; it's that parallelization matters. This provides the motivation to parallelize the structure of ParNets - with some math to perform, the way to improve performance when the clock speed cannot be increased is to do them at the same time using more resources. This technique applies equally to shallow and deep networks. Suggesting that non-deep networks are advantageous at the end of this section relies on the same assertion of theoretical minimum latency as discussed directly above in our responses, and claiming it here conflates low depth and high parallelism.**
> > >
> > > Response: In the previous response, we have tried to clarify the importance of depth in latency. We will clarify it in the paper further to not conflate low depth and high parallelism.
> > >
> > > **Reviewer momd: Table 6 suggests that ResNet may not be SOTA, so I would omit that qualifier. The best-performing networks in Table 6 are DenseNets; even if they are SOTA, I don't think that ParNet works "as well" given the differences in accuracies.**
> > >
> > > Response: We will omit the SOTA label for ResNet.
> > >
> > > Regarding “as well”, we want to clarify a potential misunderstanding. As stated in Line 257, ParNet does perform “as well” as vanilla DenseNet on CIFAR100 (24.62 vs 24.42 for 1.3 vs 1 M parameters; 20.02 vs 20.20 for 15.5 vs 7 M parameters; and 18.65 vs 19.25 for 35 vs 27.2 M parameters).
> > >
> > > ParNets do not outperform DenseNet with bottleneck and compression as stated in Line 261-263.

---

> > > > ### Comment · Reviewer_momd · 2022-08-09
> > > > **Low depth does not imply low latency**
> > > >
> > > > Thank you for your continued discussion.  However, I believe you are missing my point.  Network depth alone is not directly responsible for a network's latency in any real scenario.  Thus, it is not reasonable to focus solely on network depth in order to optimize a network's latency or performance.
> > > >
> > > > **There are some issues with this calculation. It assumes that each layer is executed in one GPU cycle.**
> > > >
> > > > I have adopted this precise assumption from your introduction: "the lowest achievable latency is *d/f*, where *d* is depth of the network and *f* is the processor frequency."  (Whether or not that processor is a GPU is immaterial here.)  In what I agree is the more realistic case that each layer is not executed in one cycle, this is because the processor is limited by hardware FLOPs, or available bandwidth, or some other resource.  In this case, depth is not the sole determining factor of network latency.
> > > >
> > > > Put another way - either
> > > > 1. The processor has enough hardware FLOPs and bandwidth to achieve a 1-cycle-per-layer latency, and the network's depth is on the critical path, or
> > > > 2. The processor must re-use its hardware FLOPs multiple times per layer or they sit idle waiting for data to operate on, at which point the network's latency is equally dependent on the number of FLOPs to be performed or the amount of data to be moved.
> > > >
> > > > I assumed the former case when concluding the difference in latencies is negligible for reasonable network depths and clock speeds, since, as you agreed, focusing on network depth as the driver of low latency only makes sense when the hardware is unencumbered by FLOPs or bandwidth.  If I've made the wrong assumption, and the second case is the one you care about (which you suggest may be the case with your example of a GPU having x/10 cores), then the network's FLOPs and memory requirements are equally (or more) important to the overall latency.  In your example, reducing the operations in each layer and reducing the number of times each cores is used will give just as much benefit as reducing the depth.  As such, there is no "depth and latency trade off" without also considering the number of FLOPs to be performed, bytes to be moved, etc.  Reducing depth but increasing width (in channels, independent streams, etc.) is not inherently beneficial.
> > > >
> > > > **For example, the control system for Fusion Reactors operates at 10KHz, or 0.1ms [1]. Such constraints restrict the neural network to be small and only of depth 4.**
> > > >
> > > > These constraints restrict the network's *complexity*, not strictly its depth.  The authors note that, to achieve a runtime in this tight latency target, they "remove superfluous weights and computations … [and] tailored the neural network structure to optimize the use of the processor’s cache and enable vectorized instructions for optimal performance." This supports my position that other factors, not just depth, are crucial to good performance.

---

> > > > > ### Author Response · Authors · 2022-08-09
> > > > > **Clarification**
> > > > >
> > > > > We believe that the reviewer is missing the point that as we have hardware with more cores, depth will increasingly become an important and limiting factor as there is no way to circumvent depth (or number of sequential steps).
> > > > >
> > > > > Also, we pointed out that the differences caused in latency by large and small depth might become a factor as we move towards applications with stringent latency requirements.
> > > > >
> > > > > In our opinion, it would be myopic to ignore the importance of depth in the mentioned example [1]. Saying that depth is important does not mean everything else, including optimized software, hardware and architecture is not important. We have many papers exploring the other factors and fewer focusing on depth, and hence our work is valuable, to point out this important dimension.

---

### Official Review · Reviewer_Zr8N · 2022-07-12

**Rating:** 5
**Confidence:** 4
**Soundness:** 3 good
**Presentation:** 3 good
**Contribution:** 2 fair

**Summary:**

In this paper, the authors explore how to reduce the depth of neural networks while keeping the model capacity. They propose ParNet, which has a depth of only 12 and can achieve comparable performance with deep neural networks. The authors also show some potential advantages of using non-deep networks for a smaller latency on future hardwares.

**Questions:**

- From my perspective, ParNet is generally comparing to models with a large computing budget like ResNet. It is interesting to see whether ParNet architecture can also achieve comparable performance with modern compact models (for example EfficientNet). Compat models are more widely used in environment that requires low-latency and some of the design philosophy might help to reduce the high computation cost of ParNet.

**Limitations:**

The main limitation of this work (inference speed on current hardwares) is well-discussed in the paper. In my option, figuring out a way to avoid the use of global average pooling and sigmoid while keeping the same depth/performance would be a huge technical contribution, as they are indeed slow on GPUs and hard to parallize or optimize.

**Strengths And Weaknesses:**

Strengths:
- The whole idea of non-deep networks is novel and not well-explored in previous works, as increasing the depth is usually a common practice for most neural network architectures to scale up themselves.
- Achieving similar performance as 'deep networks' with a depth of 12 is pretty impressive.
- The authors also discuss the potential advantages of using shallow networks to fully utilize the parallelism of hardware (e.g. GPUs).
- Some previous works are reproduced under same setting for fair comparison.
- The key limitations are well discussed.
- The writing is good and all details are presented very clearly.

Weaknesses:
- The technical contributions are not significant, as most of the crucial components of ParNet were presented in other papers. For example, the overall ParNet structure looks very similar to HRNet [1] and the ParNet block looks like a variant of Inception [2].
- Although the authors intend to improve the inference speed by reducing the network depth, ParNet still introduces some element-wise operations that are not parallism-friendly and hard to optimize (e.g. global average pooling and sigmoid in SSE layer and SiLU activation).
- To reduce the network depth, ParNet sacrifces too much on the number of parameters and FLOPs (as well as inference memory usage even it's not shown in the paper).

[1] Hu, Jie, Li Shen, and Gang Sun. "Squeeze-and-excitation networks." Proceedings of the IEEE conference on computer vision and pattern recognition. 2018.
[2] Szegedy, Christian, et al. "Going deeper with convolutions." Proceedings of the IEEE conference on computer vision and pattern recognition. 2015.

---

> ### Author Response · Authors · 2022-08-02
> **Response to Reviewer Zr8N**
>
> We thank the reviewer for positive comments and helpful feedback on our work. We address the concerns below:
>
> **Concern: The technical contributions are not significant, as most of the crucial components of ParNet are present in literature**
>
> While components of ParNet have appeared in prior literature as stated in L44, our significant technical contribution is not in proposing new components, but in using them properly for building non-deep networks. For example, parallel branches have been used in HRNet, but unlike us, HRNet introduces many interconnections between branches which reduces the degree of parallelization. Similarly, the ParNet block combines ideas from multiple sources, including Inception, RepVGG, and Squeeze and Excitation.
>
> **Concern: ParNet introduces some element-wise operations that are not parallelism-friendly and hard to optimize**
>
> We agree that ParNet introduces some operations which are currently not as optimized for parallelism as 3x3 convolution or ReLU. However, this is not a theoretical limitation, and with better software implementation the gap could be reduced. For example, global average pooling can be implemented as a parallel global mean reduce operation. Also, support for activations like SiLU and sigmoid have been improving and they are becoming as fast as ReLU [1]. We believe that if these layers are shown to be useful, better software and hardware to support them will follow.
>
> [1] https://benjaminwarner.dev/2021/07/19/benchmarking-pytorch-native-mish
>
> **Concern: To reduce the network depth, ParNet sacrifices too much on the number of parameters and FLOPs.**
>
> ParNet uses more parameters and FLOPs than deeper networks such as ResNet. However, this tradeoff may be worthwhile, if the application requires low-latency. Hence depending on the latency requirements, one might prefer a model with more parameters and FLOPs. We agree that for some applications one might wish to minimize parameters and FLOPs in particular, because of memory or energy constraints, but in this paper we explore a different objective in neural network design.
>
> **Concern: ParNet is compared to models with a high compute budget like ResNet and not compact models like EfficientNet. Compat models are more widely used for low latency and some of their design philosophy might help to reduce the compute cost of ParNet.**
>
> Thank you so much for the great suggestion! Our design philosophy is complementary to those of compact models such as EfficientNet. Combining the two might help in building low-latency and compact networks. The focus of the presented work is in showing how to build non-deep networks that achieve surprisingly high performance on vision benchmarks. Extending non-deep networks to compact design is a great direction for future work.

---

### Author Response · Authors · 2022-08-02
**Response to all reviewers**

We would like to thank the reviewers for their feedback and help in improving our work. We are excited by the support of the reviewers! We are happy that they found our work novel (Zr8N), intriguing (momd), boldly challenging conventional wisdom (momd) and well motivated (t5as, zBDf); our results impressive (Zr8N), valuable for the literature (cvGo); our experiments being numerous (t5as) and covering a wide breadth (momd); and our paper well-written and clearly presented (Zr8N, momd, t5as). Below we have addressed the individual concerns of the reviewers.

---

### Meta-Review · Area_Chair_ssTP · 2022-08-26

**Recommendation:** Accept
**Confidence:** Less certain

**Metareview:**

This work considers the task of training state-of-the-art CNNs with limited depth. The benefits considered in this work are related to the potential parallelization which is induced by depth reduction. This paper generated a fair bit of discussion with the reviewers about the motivations and the basic thesis. The authors do a good job of representing their viewpoint, and adding a version of this discussion to the final manuscript will undoubtedly be needed. The empirical results look quite promising, but the authors are also encouraged to further discuss adding an additional motivation to reducing depth (e.g., a theoretical reason, as proposed by reviewer zBDf, which is currently only one paragraph long in the related section) and/or performing a deeper study of hyper-parameters affecting accuracy/latency. With proper framing of the question studied here, the scope of evaluation and the assumptions on the hardware, this will be an interesting contribution to NeurIPS

**Award:**

No

---

### Decision · Program_Chairs · 2022-09-14

Accept